# Micro-Fragmented Adipose Tissue as a Natural Scaffold for Targeted Drug Delivery in Brain Cancer

**DOI:** 10.3390/ijms241411530

**Published:** 2023-07-16

**Authors:** Alex Salagean, Adela Nechifor-Boila, Nosherwan Bajwa, Ylenia Pastorello, Mark Slevin

**Affiliations:** 1Faculty of Medicine, George Emil Palade University of Medicine, Pharmacy, Science and Technology, 540142 Târgu Mures, Romania; salageanalex18@yahoo.com; 2Department of Histology, George Emil Palade University of Medicine, Pharmacy, Science and Technology, 540142 Târgu Mures, Romania; adelanechifor@gmail.com; 3Department of Neurosurgery, DIAKO Hospital, 24939 Flensburg, Germany; nosherwanbajwa@gmail.com; 4Department of Anatomy and Embryology, George Emil Palade University of Medicine, Pharmacy, Science and Technology, 540142 Târgu Mures, Romania; ylenia.pastorello@gmail.com; 5Center for Advanced Medical and Pharmaceutical Research (CCAMF), George Emil Palade University of Medicine, Pharmacy, Science and Technology, 540142 Târgu Mures, Romania

**Keywords:** MSC, MFAT, biomaterials, neurological cancer

## Abstract

Major limitations in the effective treatment of neurological cancer include systemic cytotoxicity of chemotherapy, inaccessibility, and inoperability. The capability to successfully target a drug to the tumor site(s) without incurring serious side effects—especially in the case of aggressive tumors, such as glioblastoma and neuroblastoma—would represent a significant breakthrough in therapy. Orthotopic systems, capable of storing and releasing proteins over a prolonged period at the site of a tumor, that utilize nanoparticles, liposomes, and hydrogels have been proposed. One candidate for drug delivery is Micro-Fragmented Adipose Tissue (MFAT). Easily obtained from the patient by abdominal subcutaneous liposuction (autologous), and with a high content of Mesenchymal Stem Cells (MSCs), mechanically derived nanofat is a natural tissue graft with a structural scaffold organization. It has a well-preserved stromal vascular fraction and a prolonged capacity to secrete anti-tumorigenic concentrations of pre-absorbed chemotherapeutics within extracellular vesicles. This review discusses current evidence supporting the potential of drug-modified MFAT for the treatment of neurological cancer with respect to recent preclinical and in vitro studies. Possible limitations and future perspectives are considered.

## 1. Introduction

Human Micro-Fragmented Adipose Tissue (MFAT) has been intensively studied in the last few years in a wide variety of research fields, showing great potential regarding its anti-inflammatory, anti-proliferative, anti-apoptotic, pro-regenerative, and, most recently, its drug delivery capabilities. Most of these properties of MFAT are associated with the presence of Mesenchymal Stem Cells (MSCs) found within the fat. MFAT is produced mechanically as liposuction-derived fat, using an enclosed sterile micronizer, such as the Lipogems or Lipocell systems. It consists of a heterogeneous mixture of adipocytes, MSCs (pericytes), endothelial cells (EC), cells belonging to the immune system, and connective tissue cells such as fibroblasts, all of which are supported by an extracellular matrix (ECM) composed of collagen fibers, polysaccharides (glycosaminoglycans), and other proteins [1,2,3].

The clinical applications in which adipose-tissue-derived MSCs, and/or MFAT, have demonstrated effectiveness include various surgical specialties [4,5]. For example, menopausal women suffering from lichen sclerosis became asymptomatic concomitant with complete epithelial regeneration of the vulva, over a period of more than a year after multiple subdermal targeted injections of autologous MFAT [6]. Whilst, in a case study, a 35-year-old woman was successfully treated for vesicouterine fistula using MFAT injection together with endoscopic repair, resulting in complete resolution of the pathology [7].

MFAT has proven effective in the treatment of perianal fistulas with Crohn’s disease. Laureti et al. treated 15 patients with persistent complex fistulizing perianal Crohn’s disease. Twenty-four weeks after a single targeted injection of MFAT, 10 of the patients had complete clinical and radiographic remission with no adverse events reported, demonstrating that this could be a novel and potentially effective minimally invasive treatment for inflammatory bowel disease [8].

MFAT has been used most frequently in the field of orthopedic surgery, with promising published clinical studies in the treatment of joint injuries and osteoarthritis (OA) in animals (e.g., racehorses and dogs) and humans. For example, Giorgini et al. treated 49 patients with moderate to severe knee OA using a combination of arthroscopy and a single injection of MFAT. In this 2-year retrospective single-center study, long-term significant improvement as measured with the knee injury and OA outcome score (KOOS) was shown [9].

These and other studies demonstrate that the use of autologous MFAT injection is an innovative and safe method for treating inflammatory-based disease [5,10,11,12]. This procedure is characterized by its simplicity, affordability, speed, minimal invasiveness, one-step application, low risk of complications, and compliance with regulatory guidelines [4].

In addition, recent publications have found that MFAT, devitalized-MFAT (D-MFAT), and MSCs alone, can all absorb and release chemotherapeutic drugs such as paclitaxel (PTX) in dose-dependent and potentially anti-tumorigenic quantities, over periods of several days (MSC) to several months (MFAT) with their secretome, previously shown to be capable of inhibiting the growth and development of cancer cell lines such as U-87 MG (glioblastoma); CFPAC-1(human pancreatic adenocarcinoma); IMR-32, NB-1619 (wild type and luciferase transfected neuroblastoma), as well as tumor growth in vivo (discussed later in more detail) [13,14].

Glioblastoma, the most common brain tumor (annual incidence of more than 5 in 100,000), is a difficult tumor to treat because of its location and its integration within the host parenchyma, making it almost impossible to completely remove surgically [15]. The prognosis is poor with a median survival of only 2 years after diagnosis, and treatment options remained the same as they were a decade ago—involving surgery, where possible, followed by temozolomide with concurrent radiotherapy. More recently, the antiangiogenic drug bevacizumab (Avastin) has been tested, but clinical trials did not show any improvement in survival time [16]. The use of current therapies to treat glioblastoma more effectively is limited by the unique and hypersensitive microenvironment of the brain, together with difficulties in achieving sufficient drug transfer across the blood–brain barrier (BBB), and in addition, treatment is associated with significant system cytotoxicity [17]. Therefore, novel targeted approaches involving sustained de novo delivery of chemotherapeutics could improve morbidity and reduce mortality.

## 2. Anti-Inflammatory and Other Anti-Cancer Beneficial Properties of MFAT

The protective effects of MFAT revolve, in part, around its remarkable anti-inflammatory capacity deriving primarily from the release of cytokines from the MSC. The MFAT tissue graft can display organ-protective properties as shown by the improvement in survival rates in sepsis models. For example, the administration of human MFAT significantly improved the inflammation score and survival rate in a cecal ligation murine model of severe sepsis. This beneficial effect was mediated through the activation of COX-2 and resulted in a significant reduction in the expression of pro-inflammatory cytokines including Interleukin-6 (IL-6) and Interleukin-1 beta (IL-1β) [18].

### 2.1. MFAT Extraction and Characterization

MFAT is prepared from abdominal adipose tissue obtained through liposuction using an enclosed sterile process, involving micronization to nanofragments of around 0.3–1 mm in diameter and cleaning to remove excess oil, haematological cells, and fibrous tissue. The most commonly used kit is called LipoGems S.p.A. Firstly, a mini liposuction is carried out in the abdominal region following an injection of Klein solution in a locally anesthetized donor, and approximately 100–200 mL of fat is removed. Next, the fat is ‘sieved’ with a filter set (0.7 mm diameter holes) and homogenized in an enclosed sterile tube, held under low pressure in the presence of ball bearings, and finally washed in a large volume of saline, resulting in the production of approximately 10 mL of MFAT per 100 mL of original extract. The processed MFAT consists primarily of clusters of perivascular cells held together in the form of micrografts with adipocytes, which can be collected in a Luer Lock syringe and injected locally where needed [3,19].

MFAT clusters can be cultured directly in a tissue culture medium in order to assess their size, viability, and activity. In addition, MSCs can be isolated following collagenase-I digestion, with agitation over a period of 1 h. The capacity of MSCs to form colony-forming units (CFU), and the level of secretion of cytokines and growth factors, is indicative of their overall activity and that of the parent MFAT [20].

Vezzani et al. analyzed the composition and secretome of MFAT, and they compared it with that of enzymatically digested stromal vascular fraction. They showed a significantly higher concentration of pericytes (MSCs) and a release of growth factors and cytokines associated with tissue regeneration and repair, including angiopoietin, tissue inhibitor of matrix metalloproteinase (TIMP), and platelet factor-4 (PLT-4) [21]. Further studies confirmed the critical presence of conserved stromal structural components and extracellular matrix proteins, such as collagen type 1, with a functional microvascular presence together with MSCs (confirmed by their capability to form colony-forming units and expression of stem cell surface markers including SOX2, NANOG, and OCT3/4). In addition, their exosome-derived secretome was shown to contain complex miRNAs and proteins protecting against macrophage M1 polarization, and therefore, inflammation [22,23]. In addition to the biological properties, MFAT has important biomechanical functions. When used as a graft, it provides a supportive matrix that imparts cushioning, lubrication, and protection, for example, when injected into the joint space in people with OA [2].

### 2.2. MSC Paracrine Effects

MSCs play a crucial role in tissue regeneration and act as medicinal signaling cells. These cells can be obtained from various sources like bone marrow, placenta, umbilical cord blood, and adipose tissue. MSCs and pericytes are effectively the same cells, exhibiting comparable marker expressions and, more importantly, demonstrating similar functional characteristics. Pericytes form the smooth muscle cell coating of microvessels in the fat and, during its activation, the pericytes detach from the surface and become MSCs. It is worth noting that both cell types are considered safe for allogeneic transplantation due to their lack of expression of immune-related, membrane-bound molecules [24].

MSCs secrete a diverse array of bioactive molecules that function in a paracrine manner. These molecules play a crucial role in priming and sustaining angiogenic, anti-fibrotic, anti-apoptotic, and immunomodulatory responses within the target tissue. Maximal therapeutic benefit is conferred, since subcutaneous fat represents the tissue with the highest concentration of MSCs, being far superior to bone-marrow-derived sources [3]. MFAT-derived MSCs produce colony-forming units indicating their stemness. In addition, they secrete substantial quantities of the critical anti-inflammatory regulating molecule Interleukin-1 receptor alpha antagonist (IL-1Rαa), pro-regenerative hepatocyte growth factor (HGF), angiogenic transforming growth factor betas 1 and 2 (TGFβ1/2), and anti-bacterial chemokine C-X-C motif-ligand-9 (CXCL-9), as well as a multitude of other growth factors and cytokines [20].

#### 2.2.1. MSCs and Angiogenesis

Published data have shown that MSCs have the potential to mitigate the extent of cerebral infarction following ischemia and contribute to functional restoration. One proposed mechanism is that MSC transplantation after a stroke enhances angiogenesis, by either producing or amplifying endogenous factors essential for blood vessel formation. These factors include vascular endothelial growth factor (VEGF), angiopoietin-1 (Ang-1), placental growth factor (PlGF), and fibroblast growth factor-2 (FGF-2) [25].

The presence of these growth factors can support the development and the effective maturation of vascular structures associated with revascularization, ultimately—for example—in an ischaemic stroke, MSC treatment can result in a possible reduction of the size of the infarcted area. Asgari Taei et al. delivered an embryonic MSC-derived conditioned medium, intracerebroventricularly, to male Wistar rats following middle cerebral artery occlusion and showed increased expression of angiogenic markers, including CD31 in affected cortical regions, with the concomitant reduction in neurological deficits and infarct volume [26]. With regard to tumor angiogenesis, MSC paracrine modification of the microenvironment within the tumor should help to normalize the vascularization process, producing more patent microvessels capable of delivering the targeted therapeutic more effectively throughout the stroma of the tissue.

MSCs derived from bone marrow were engineered to express kringle-5 (an angiogenesis inhibitor from human plasminogen) under the control of early growth factor-1. Systemic intravenous administration of this modified potential therapeutic resulted in a reduction in tumor growth, and it improved survival in a murine glioblastoma xenograft model, suggesting that MSC-based therapies could act in a dual role, blocking tumor vascularization through direct paracrine secretion and through drug delivery strategies [27].

#### 2.2.2. Neuroprotective Effects of MSCs

A study conducted on experimental animals transplanted human umbilical cord-MSCs (hUC-MSCs) into neonatal rats, resulting in reduced tissue damage and infarct volume due to the migration of cells into the periventricular tissue space. This treatment also led to improved motor function in the neonatal rats. In addition, hUC-MSCs demonstrated significant reductions in apoptosis as well as the expression of beclin-2 and caspase-3, which are critical regulators of the apoptotic cascade [28].

Other research further demonstrated that the anti-apoptotic effects of hUC-MSCs involve the Bcl-2 pathway, which provides neuroprotection against ischemic stroke when administered in low doses. Numerous other studies have reported the beneficial effects of MSC-based therapies in promoting functional restoration in cases of hypoxic–ischemic brain damage by exerting immunomodulatory effects, as eloquently described in the review by Isaković et al. [29]. It should be remembered that MSCs do not possess the ability to differentiate into neurons and neuroglia in vivo, meaning that they would not contribute directly to regenerative tissue restoration [30].

#### 2.2.3. Evidence for the Role of MSCs in Tumor Modulation

Considerations must be given to the potential stimulatory effect of MSC-derived growth factors on angiogenesis-dependent glial tumor growth. Angiogenesis is of critical importance for the optimal proliferation and expansion of glioblastoma [31]. A meta-analysis of clinical trials involving the use of angiogenesis inhibitors showed that only bevacizumab, given as a single treatment, resulted in an improved response to temozolomide, with a significant median progression-free survival at 6 months but no overall improvement in survival time, indicating the relevance but limitation of neo-vascular development to the tumor growth [32]. Nowak et al. showed that MSC treatment could exert anti-tumoral activity through regulation of both apoptosis and vascular development [33]. Since it is now known that normalization of tumor vasculature can improve the delivery of chemotherapeutics, further research should be carried out on the subject.

Nakamizo et al. demonstrated through in vivo and in vitro experiments that MSCs derived from bone marrow can migrate to tumor sites and, when injected adjacent to glioma xenografts, could home effectively to the tumor (in an HGF-dependent mechanism), interact and integrate with glioma cells, and inhibit their growth [34,35]. Therefore, MSCs have the potential to enhance the effectiveness of conventional glioma treatments, such as chemotherapy and radiation therapy, by reducing treatment-induced side effects, modifying the tumor microenvironment, and improving therapeutic outcomes [34]. In this regard, exosomes derived from rat bone marrow, which deliver the ‘payload’ from the MSCs, were shown to induce C6 glioblastoma cell apoptosis in vitro in an Akt, caspase 3-dependent fashion [36].

#### 2.2.4. MSC-Derived Exosomes as Cell Free Targeting Therapeutics

Exosomes are microvesicles released from MSCs that contain all the active proteins which contribute to immunomodulation and cellular regeneration. For this reason, attention has been given to their use in therapy due to their smaller size (allowing penetration through the BBB) and allogeneic capacity. In addition, they retain the encapsulated nature and half-life of the parent cell without the accompanying immuno-compromising characteristics (HLA antigen expression) [37]. Khayambashi et al. eloquently described how hydrogel encapsulation of exosomes could be used as a method to further compliment the controlled extended release of paracrine factors in vivo [38]. Studies have shown that either a local or systemic delivery of exosomes can successfully dampen the immune response to tissue injury and disease in vivo, through both protein and miRNA effects on macrophages, T regulatory cells, and dendritic cells, concomitantly reducing oxidative stress, activating autophagy, and inhibiting apoptosis in the brain parenchyma [39].

Do et al. contemplated the use of exosomes derived from MSCs as a mechanism for targeted delivery in brain tumor treatment. The advantages of exosomes over the parent MSCs include preferential accumulation in the brain parenchyma (homing), an improved safety profile (lower risk of induction of tumor growth), and a longer systemic half-life. Whilst direct parenchymal injection resulted in greater incorporation into the brain tissue than with MSCs. They retain the capacity to uptake and deliver therapeutic drugs and can be targeted, for example, via chemical modification to glioblastoma (neuropilin-1) [40]. Chistiakova et al. demonstrated effective inhibition of glioblastoma cell growth in vitro using an MSC-derived exosomal conditioned medium (see the review of Do et al. for further in vitro and in vivo evidence [40]) [41]. In their review, Lin et al. discussed evidence supporting the phenomenon that exosomes derived from MSCs enhance chemosensitivity and reduce resistance of breast cancer cells exposed to cisplatin. Also, miR-451a completely blocked epithelial mesenchymal transition (EMT) in hepatocellular carcinoma and other cell lines maintaining the cells’ sensitivity to PTX [42]. In summary, the properties of exosomes, including tumor tropism, half-life, BBB permeability, natural intracellular communication, and parenchymal fusion, suggest their inclusion as an ‘adjuvant’ within MFAT, further optimizing its potential antitumor effects, particularly in the treatment of neurological cancers.

#### 2.2.5. Adipocyte Activity from MFAT

Adipocytes themselves possess strong anti-inflammatory properties that contribute significantly to the immunomodulatory function of MFAT. More specifically, adiponectin is a highly secretive immunoprotective molecule that synergizes with MSC-secreted cytokines to protect the balance of the local microenvironment [43]. Hence, the fat cells create the crucial scaffolding for ensuring graft survival in vivo. In addition, since they constitute 50% of the total cellular component of MFAT (as compared with MSCs which account for only 1% of the content), their role in supporting the anti-inflammatory status of the microenvironment implant site is of significant importance [44].

#### 2.2.6. Uptake of Drugs and Engineered Delivery Vehicles

MSCs can also serve as therapeutic carriers and have recently been shown to absorb chemotherapeutic drugs, releasing them in microvesicles at a regular, useful physiological concentration for up to several months.

The intrinsic ability of MSCs to migrate towards tumor sites makes them promising vehicles for the delivery of pharmacological agents. This approach has shown promise in the targeted delivery of oncolytic viruses as potential anticancer therapies and has potential applications in other cancer-related treatments [45]. For example, Mangraviti et al. incorporated genetically engineered Bone Morphogenetic Protein-4 (BMP-4) plasmid DNA containing nanoparticles inside adipose-tissue-derived MSCs. Following either intranasal or systemic IV delivery, the MSCs homed effectively to glioma and tumor cells, penetrating the lesion and releasing BMP-4, resulting in improved survival in a rat model [46].

Bonomi et al. first showed in 2017 that MSCs derived from either bone marrow or adipose tissue loaded with PTX could inhibit proliferation of human glioblastoma cell lines T98G and U87MG over a period of 7 days, concomitant with release of the drug into the tissue culture media [47]. Pacioni et al. demonstrated effective homing and reduction in tumor growth of rat glioblastoma xenografts (U87MG), following IV or IA injection of MSCs, providing evidence that the native cells alone could be anti-tumorigenic [48]. MSCs or their secretome were able to inhibit mesothelioma cell growth in vitro and mesothelioma cell murine xenografts in vivo, with an efficiency similar to that produced with an IV injection of PTX [49]. Combining the above concepts, Coccè et al. successfully abrogated U87MG growth in vitro using adipose-derived MSCs engineered to contain tumor necrosis factor-related apoptosis-inducing ligand protein (TRAIL) and PTX, suggesting multimodal MSC therapies might be more efficacious [50].

These findings suggest that human MSCs hold promise as a potential therapeutic strategy for gliomas.

## 3. Proof-of-Concept Evidence for Effective Drug Delivery Using MFAT

Alessandri et al. highlighted that both MFAT and its D-MFAT derivative exhibited comparable and effective abilities to incorporate PTX. Both substances rapidly assimilated PTX, with 5 min being sufficient for them to incorporate 90% to 95% of the chemotherapeutic (2 µg/mL), following the priming of 24 exposures to PTX stock solution (6 mg/mL). The majority of PTX was localized within their adipocytes/lipid content and is dependent upon interaction between the drug and hydrophobic nature of fat. Moreover, they both displayed a slow and efficient release of active PTX in its original form in vitro and could be re-loaded with drugs multiple times without loss of potency and without any other metabolites present. This release persisted for at least 2 months, with D-MFAT releasing double the amount of PTX compared to MFAT [13]. The exact mechanism through which MFAT allows the consistent, slow release of chemotherapeutics, and probably other drugs, is still unknown.

Due to its presentation of spontaneous diseases closely resembling human oncology, the domestic dog is regarded as a valuable animal model for assessing novel drugs and therapeutic approaches. In this regard, a case reported the efficiency of MFAT primed with PTX in a 6-year-old dog diagnosed with mesothelioma. Here, Zeira et al. treated a neutered mixed-breed dog weighing 24 kg, which exhibited progressive weakness, loss of appetite, productive cough, abdominal distension, and breathing difficulties for a period of 2 months. In this case, autologous adipose tissue was obtained through lipo-aspiration from the lumbar flanks of the dog. The adipose tissue was then micro-fragmented using a specialized device that allows for minimal manipulation without the need for enzymatic procedures. The micro-fragmented samples were used as a scaffold for PTX (MFAT–PTX). The treatment protocol involved administering 7 mL of MFAT–PTX into the abdominal and thoracic cavity under ultrasound guidance. Over a period of 22 months, the dog underwent a total of 17 treatments, with both intrathoracic and intra-abdominal administrations. On average, the treatments were administered every 38 days, with the shortest interval being 14 days and the longest being 70 days. In this case report, the treatment approach using MFAT–PTX appeared to have the ability to generate a local, sustained antineoplastic effect without causing any systemic myelotoxicity. Also, no complications regarding the treatment were stated in the case report [51].

Both MFAT and D-MFAT exhibit properties of natural biological scaffolds, capable of absorbing and releasing a substantial amount of drugs such as PTX. In an orthotopic murine model of human neuroblastoma (HTLA-230), the local administration of D-MFAT–PTX at the tumor site (left adrenal gland), following its surgical resection, resulted in the blocking of cancer relapse, whilst treatment with PTX alone only delayed the recurrence up to 22 days. Pharmacokinetic studies revealed that D-MFAT-delivered PTX (subcutaneously) resulted in a high local concentration, significantly reducing the systemic concentration of PTX compared to systemically administered D-MFAT–PTX (intraperitoneally). These findings strongly support that MFAT can serve as a natural biomaterial capable of absorbing and transporting chemotherapeutic drugs. Unexpectedly, even the devitalized derivative (D-MFAT) remained effective, indicating that the presence of viable MSCs or adipocytes in MFAT was not essential for the uptake, release, and anti-cancer effect of the drug [13].

However, the presence of either viable or non-viable adipocytes/lipid content in both MFAT and D-MFAT specimens appeared crucial for facilitating the rapid uptake of PTX at physiologically useful concentrations. Based on these findings, we have concluded that both MFAT and D-MFAT have the potential to serve as innovative natural biomaterials, facilitating the localization and controlled release of anti-cancer compounds at the tumor site. These findings offer promising perspectives for their potential application in human cancers [13].

## 4. Discussion

The use of MSCs in clinical applications has been utilized in hundreds of clinical trials; however, almost all of them have failed, probably due to heterogeneity of sample sources, number of cells used in the treatment, and the method of delivery (IV, IA, intrathecal, etc.) Furthermore, stem cells cannot survive for more than about 3 days in circulatory systems, and considering that in the form of cell therapy they do not incorporate directly into damaged or diseased tissue, this indicates a short potential half-life and positive effect on the target tissue microenvironment. Although currently there are over a thousand ongoing trials, and many others completed, the only stem-cell-based treatment that is routinely reviewed and approved by the U.S. Food and Drug Administration (FDA) is hematopoietic (or blood) stem cell transplantation [52]. Therefore, more considered approaches should be made to harness the power of the paracrine secretions obtaining a longer lasting targeted impact on the tissue of interest, at the same time utilizing the ability of a natural scaffold such as MFAT to deliver a combination of therapeutics.

The gold standard treatment of glioblastoma is multimodal, involving surgery whenever possible, chemotherapy with temozolomide, or a combination of temozolomide and radiotherapy. The average survival time, despite the treatment, is only 15 months [53]. The administration of tumor treatment drugs often leads to harmful side effects in patients. This is primarily due to the drugs’ systemic toxicity, as the intravenous injection of chemotherapeutics affects multiple targets within the body.

Extensive research is being conducted to explore approaches that minimize drug side effects while increasing therapeutic concentrations, incorporating new drug delivery systems specifically at the tumor site. This has been the focus of this review, with MFAT being the most promising natural biological scaffold to date (Figure 1).

Evidence suggests that carefully selected targeted injection of nano- or micro-fat into the brain could be relatively safe, and a potential option—in the form of a clinical trial—in those people where other treatments have failed. Adipose tissue has a known capacity to be able to absorb and retain even highly toxic chemicals or drugs, whilst retaining its cellular viability and function, making it ideal for storing chemotherapeutics prior to targeted delivery.

Alessandri et al. devitalized MFAT (D-MFAT), derived from LipoGems devices, using several freeze-thaw cycles (−20 °C), resulting in the destruction of the stromal vascular fraction. They proceeded to demonstrate that D-MFAT retained the original architecture of the viable microfat and was equally as effective as MFAT in absorption and release of chemotherapeutic drugs, whilst potentially being safer to use as a therapeutic [13].

Current limitations include the lack of homogeneity and replicable preparation of MFAT. The cellular components of the graft vary in activity and concentration from sample to sample, and person to person, making it difficult to define a statistically meaningful result from the so far relatively small clinical studies. On the other hand, MFAT is currently FDA-cleared for human clinical use, and following optimization for combinational drug uptake and delivery, could be admitted for human trials. Exosomes may also be incorporated back into MFAT in order to boost the concentration of the active secretome or could be considered separately, either alone or encased within synthetic hydrogels that could be safer when delivered into the brain.

## 5. Conclusions

MFAT represents an incredible natural scaffold that has the capacity to deliver targeted combination sets of drugs over a concerted period of time, supporting the more positive treatment outcomes in conditions such as glioblastoma, which occurs in surgically challenging anatomical locations. In addition, the high concentration of MSCs within the tissue graft provides both anti-tumoral activity and positive anti-inflammatory immunomodulation. Limitations may include the requirement in chronic disease for multiple semi-invasive graft injections, although MFAT produced from liposuction can be effectively cryopreserved for future use. Secondly, we should further consider any issue associated with the delivery and fate of adipose tissue within the central nervous system (CNS).

## 6. Future Directions

Further studies should investigate and optimize the capability of MFAT for drug delivery, focusing on specific combination glioblastoma chemotherapeutics (where PTX plus temozolomide appears to have a synergistic beneficial effect). Also, the graft and cluster size of MFAT should be characterized to maximize drug release. The optimal size of the graft is approximately 0.8 mm; however, current preparations are very heterogenous, meaning that delivery concentration, as well as graft viability, are suboptimal. Finally, the constituent MSCs and adipocytes of the graft also vary in number and activity from patient to patient, and this should be addressed in order to enable a consistent future clinical application, particularly considering the crucial paracrine role that these ‘medicinal cells’ partake in the immunomodulation, protection, and regeneration of diseased tissue.

## Figures and Tables

**Figure 1 ijms-24-11530-f001:**
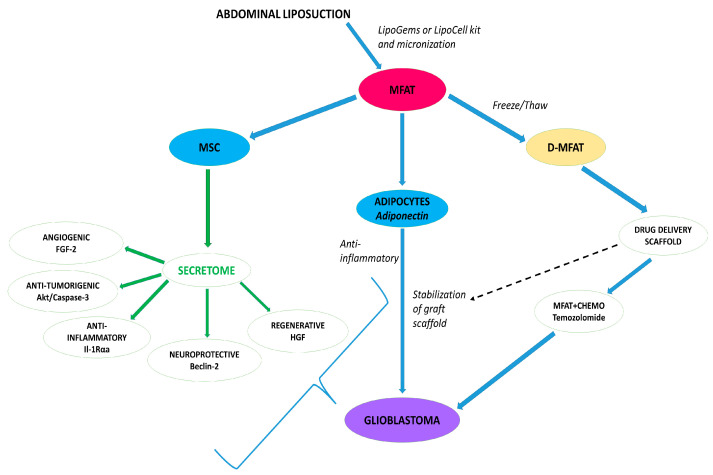
Autologous MFAT is effectively and homogenously produced following liposuction using, e.g., LipoGems SPA kit (with a combination of glass marbles filters and saline washing). The figure highlights critical components of the micro-graft including active cellular proteins released as the secretome within continuously released exosomes, primarily from MSCs. The adipocyte-bound-stable scaffold has the capacity to absorb chemotherapeutic drugs, such as temozolomide, and release them in the form of a graft targeted to the tumor over a period of up to several months. With an appropriate catheter in place, multiple therapies could be given that protect against tumor growth indefinitely. AKT, Protein Kinase B; D-MFAT, Devitilized-Micro-fragmented Adipose Tissue; FGF-2, Fibroblast Growth Factor-2; HGH, Hepatocyte Growth Factor; IL-1Rαa, Interleukin-1 Receptor Alpha a; MSC, Mesenchymal Stem Cells; MFAT, Micro-fragmented Adipose Tissue.

## Data Availability

Not applicable.

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
