# Peer review of "Micro-Fragmented Adipose Tissue as a Natural Scaffold for Targeted Drug Delivery in Brain Cancer"

_ijms, 2023, doi:10.3390/ijms241411530_

Round 1
Reviewer 1 Report
This is an interesting review on a subject which deserves more coverage in scientific journals since the concept of using MSC's & MFAT for tumour therapy is novel, intriguing and I recommend acceptance.
Comments:
Possibly insert a brief comment on the potential toxicity that exogenous therapeutic agents may have against MFAT derived MSC's or adipocyyes and why this is not a problem for tumour targeted therapy?
Define more explicitly, the difference between tumour-induced angiogenesis and normal angiogenesis and how MSC may positovely influence this balance> Clarify why MSC derived stimulation of normal angiogenesis does not augment tumour growth?
Minor Comments
Line 38 rephrase as "liposuction-derived fat"
The Font on page 5 is mixed.
Author Response
Possibly insert a brief comment on the potential toxicity that exogenous therapeutic agents may have against MFAT derived MSC's or adipocyyes and why this is not a problem for tumour targeted therapy?- we added a statement explaining this in the discussion (pg.8)
Define more explicitly, the difference between tumour-induced angiogenesis and normal angiogenesis and how MSC may positovely influence this balance> Clarify why MSC derived stimulation of normal angiogenesis does not augment tumour growth?- we added an explanation for this on page 4
Minor Comments
Line 38 rephrase as "liposuction-derived fat"- we rephrased this
The Font on page 5 is mixed.-it has been corrected
Reviewer 2 Report
This manuscript is written clearly and intelligibly. The experimental results are consistent with the conclusions made by the authors. I consider the content of this manuscript will be of some interest to readers. However, the authors should improve the soundness of this study.
Author Response
This manuscript is written clearly and intelligibly. The experimental results are consistent with the conclusions made by the authors. I consider the content of this manuscript will be of some interest to readers. However, the authors should improve the soundness of this study.- we have devoted time to carefully improving the manuscript. Please, see the highlighted sections.
Reviewer 3 Report
The Author should revise the review as per the following suggested points
1. add a paragraph on harvesting and Processing of the Adipose Tissue
2. add information about the Isolation and Culture of MFAT
3. add Biomechanical Properties of MFAT
4. add a discussion about devitalized MFAT (DMFAT)
5. include the additional information in detail such as cell priming, release behaviour other characterization techniques ( in vitro and invivo)
NA
Author Response
1. add a paragraph on harvesting and Processing of the Adipose Tissue- we added this. Please see page 3
2. add information about the Isolation and Culture of MFAT- this is also been included (pg.3)
3. add Biomechanical Properties of MFAT- we added this information on page 3
4. add a discussion about devitalized MFAT (DMFAT)- this was added on page 8
5. include the additional information in detail such as cell priming, release behaviour other characterization techniques ( in vitro and invivo)- added on page 6
See the blue highlighted sections in the latest manuscript draft.
Round 2
Reviewer 3 Report
Article can be accepted in the present form without any change